# Task Matrices: Linear Maps for Cross-Model Finetuning Transfer across Modalities

**Darrin O'Brien**[1]* **Dhikshith Gajulapalli**[1] **Pranay Rishi Nalem**[1] **Alexander Ramsey**[1]

**Eric Xia**[2]†

[1]Algoverse AI Research [2]Brown University

## Abstract

Results in interpretability suggest that large vision and language models develop implicit linearities in pretrained settings. Learned linear encodings have been documented in in-context learning settings, where model predictions are biased at runtime. However, it is unclear whether similar linear representations exists in more generalized adaptation regimes. In this work, we develop the concept of a task matrix, a linear transformation from a base to finetuned embedding state. We demonstrate that for CLIP, DEiT, DINOv3, allMiniLM-V2, and RoBERTa, a base model augmented with a task matrix approaches finetuned accuracies on certain datasets, while resulting in marginal improvements on others. Our results demonstrate that over a range of models, modalities, and tasks, linear encoding in transformer embedding spaces exist not only between layers in a single model architecture, but also between pretrained and finetuned architectures.

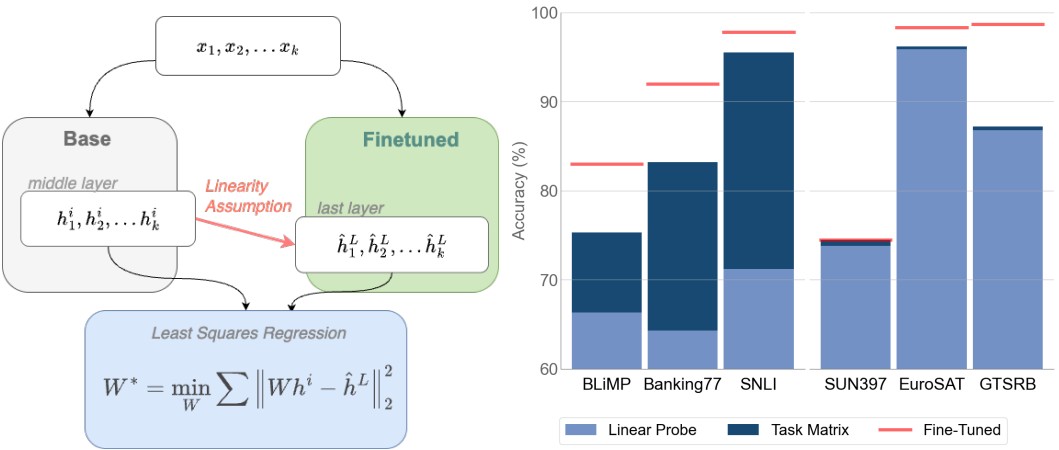

Figure 1: **Left:** On many datasets, employing a linearity assumption between base and finetuned model states offer lightweight and effective approximations. **Right:** Applying a task matrix beats linear probes, and sometimes reaches finetuned performance.

---

*Lead author

†Senior author

# 1  Introduction

Finetuning has cemented itself as the traditional approach for adapting foundation models for specific downstream tasks [Devlin et al., 2018], though at the cost of substantial training time and computation. Hence, there has recently been an increasing interest in developing lightweight alternatives to finetuning such as linear probes and low-rank adaptation (LoRA) [Alain and Bengio, 2018, Hu et al., 2021].

In this work, we employ a concept learning hypothesis to develop a novel method for transferring fine-tuned performance to base models. First introduced by Paccanaro and Hinton [2001], linear transformations between vector representations have been found to be effective for relational approximation between given concepts. In the transformer architecture setting, Hernandez et al. [2023] demonstrated that model architectures often employ near-linear transformations over relations in the setting of **in-context learning**. Consequently, based on interpretability results highlighting representational flexibility in middle layers, we introduce the concept of the **task matrix**:

---

A **task matrix** is an $N_{\text{embed}} \times N_{\text{embed}}$ linear transformation from a base model representation to a fine-tuned representation, where the finetuned model has been trained on a dataset $D$. This task matrix is built upon a **linearity assumption**. Specifically, we propose that a linear map $W$ transforms the hidden representation at a fixed intermediate layer of a base model, $x \in H_{\text{base}}$, into the last-layer representation of the finetuned model $y \in H_{\text{ft}}$:

$$Wx \approx y$$

The task matrix is then constructed through regression over samples from $D$, on pairs of base and finetuned hidden representations.

---

Multiplying base embeddings by a task matrix then produces an approximation of the finetuned output, which is passed to downstream head(s) for decoding.

We find that on the majority of datasets, tasks matrices outperform probing baselines, approach fine-tuned performance in constrained data regimes, and generalize over multiple tasks.

# 2  Related Work

Within concept learning, relationships between vector encodings have long been represented as matrix transformations, for instance in hierarchical data structures and models of compositional semantics [Paccanaro and Hinton, 2001, Coecke et al., 2010].

Subsequently, a substantial body of interpretability literature have provided evidence for linear representation of concepts within model architectures [Mikolov et al., 2013, Elhage et al., 2022, Park et al., 2024].

In the domain of transformers, linear representations has likewise been utilized to identify concepts and modify predictions through hidden representation interventions [Hernandez et al., 2023, Chanin et al., 2024, Xia and Kalita, 2025]. We take inspiration from the setup and hypothesis of these works, especially the **middle state enrichment** found by Geva et al. [2021]. However, unlike prior work, we hypothesize linear representations over domain adaptation between pretrained and finetuned models, not only under a relational constraint.

# 3  Approach

## 3.1  Preliminaries & Linearity Assumption

We focus on transformer architectures, which have been applied successfully across vision, text, and multimodal tasks. We formalize the nonlinear transformations in a transformer as mappings between successive vector spaces. Let the initial embedding be $h^0 \in \mathbb{R}^d$, where $d$ is the hidden dimension. These embeddings are updated by $L$ transformer blocks, such that for each $\ell \in [1, L]$,

$$h^{\ell} = b^{\ell}(h^{\ell-1}), \text{ where } b^{\ell} = b^{\text{ffn},\ell} \circ b^{\text{attn},\ell}$$

is a composition of multi-head self-attention and feed-forward layers, with residual connections and layer normalization. The final representation $h^L$ is then projected into a task-specific output space by a finetuned classification head.

Our linearity assumption is as follows: let the finetuned model's output space be $K^{\text{ft}}$, and the base model's output space at the $i^{\text{th}}$ layer be $K^i_{\text{base}}$. We assume that there exists some $i \in \{1, 2, \ldots, L\}$, a sample population $k$, and a matrix $W \in \mathbb{R}^{d \times k}$ such that for all pairs $(x, y) \in K^i_{\text{base}}, K^{\text{ft}}$,

$$Wx \approx y$$

## 3.2 Task Matrix Construction

Let $h^i \in \mathbb{R}^d$ and $\hat{h}^L \in \mathbb{R}^d$ represent the mid-layer embedding of a pre-finetuned model and the final-layer embedding of a finetuned model, respectively. To estimate the task matrix $W$ that maps $h^i \mapsto \hat{h}^L$, we assume a linear transformation $W$ holds between these states across inputs $x_1, x_2, \ldots, x_k$. That is, across all pairs $(h^i_k, \hat{h}^L_k)$ for a sample population $k$, we posit that a single transformation matrix W exists from an intermediate base layer to final finetuned layer state:

$$\hat{h}^L_k \approx W h^i_k$$

To approximate $W$ with a learned matrix $W^*$, we solve a least-squares regression problem which finds the linear transformation minimizing the reconstruction error across the sample population $x_1, x_2, \ldots x_k$.

$$W^* = \min_W \sum_{i=1}^{k} \left\| W h^i_k - \hat{h}^L_k \right\|_2^2$$

At inference time, for test sample $j$, we multiply $W^*$ with the intermediate representation $h^i_j$ and pass the result through the fine-tuned classification head to obtain predictions.

# 4 Methodology

Our experiments focused on architectures with sufficient depth (greater than 10 layers), as shallow networks demonstrated reduced efficacy in cross-layer transformation for approximating fine-tuned representations. We selected datasets exhibiting substantial performance gaps between base and fine-tuned models, enabling meaningful evaluation of task matrix transformations. We then constructed task matrices as outlined above; see the Appendix for further details.

## 4.1 Image Classification

For image classification, we used the CLIP ViT-B/32 (Radford et al. [2021]) Vision Tower, and trained an end-to-end classification network. In the appendix, we also show results for DeIT (Touvron et al. [2021]) and DINOv3 (Siméoni et al. [2025]), demonstrating our approach is generalizable.

We tested on the following diverse datasets: DTD (Cimpoi et al. [2014]), EuroSAT (Helber et al. [2019]), GTSRB (Stallkamp et al. [2012]), MNIST (LeCun et al. [2010]), RESISC45 (Cheng et al. [2017]), Stanford Cars (Krause et al. [2013]), SUN397 (Xiao et al. [2010]), and SVHN (Netzer et al. [2011]). The datasets encompass diverse visual classification tasks, including texture recognition, scene categorization, vehicle identification, digit classification, and traffic sign detection.

## 4.2 Text

In autoregressive language models, the effects of domain adaptation on implicit linear mappings between specific token positions have not yet been studied in depth. In order to simplify our experiments, we employed sentence transformer architectures (Reimers and Gurevych [2019] derived from BERT, specifically all-MiniLM-L12-v2 (Wang et al. [2020]), and RoBERTa-large (Liu et al. [2019]). This choice means that tokenization of sentences immediately leads to states for which the linearity assumption can be applied.

We evaluated the models across seven diverse NLP benchmarks: Emotion [Saravia et al., 2018], HANS [McCoy et al., 2019], BLiMP [Warstadt et al., 2020], TREC-6 [Li and Roth, 2002], SNLI [Bowman et al., 2015], and Banking-77 [Casanueva et al., 2020]. See the Appendix for detailed dataset descriptions.

## 5 Results

Below, we show the efficacy of task matrices at exploiting non-final layer linearities, demonstrate they are robust to data-constrained and multi-task settings, and validate their casual influence to predictions.

### 5.1 Task Matrix Performance

We find the strongest results for RoBERTa, outperforming linear probes from the same data distribution on all seven tested datasets. On the vision side, we find similar results and often come within a percentage point of finetuned accuracy, while linear probes also perform well.

Table 1: Task Matrix against text baselines (%), RoBERTa-large (n=5, 95% CI). Layers are zero-indexed.

| Method | Emotion | HANS | BLiMP | Trec-6 | SNLI | ATIS | Banking77 |
|--------|---------|------|-------|--------|------|------|-----------|
| (classes) | (6) | (2) | (67) | (6) | (3) | (18) | (77) |
| Linear Probe | 58.9±1.5 | 81.3±0.8 | 66.3±1.5 | 79.8±1.5 | 71.2±0.5 | 89.3±0.2 | 64.3±3.0 |
| Task Matrix | **66.0±2.8** | **96.8±0.2** | **75.3±1.0** | **84.9±1.2** | **76.3±1.8** | **95.5±0.3** | **83.2±1.4** |
| (best layer) | (1,2,10) | (16) | (4,5,6) | (11) | (17) | (4,6) | (1,3,4) |
| Fine-Tuned | 91.4±0.8 | 100.0±0.0 | 83.0±1.0 | 95.1±1.6 | 88.7±0.6 | 97.8±0.3 | 92.0±0.7 |

Table 2: Task Matrix against vision baselines (%), CLIP ViT-B/32 vision tower (n=5, 95% CI). Layers are zero-indexed.

| Method | DTD | EuroSAT | GTSRB | MNIST | RESISC | Stanford Cars | SUN397 | SVHN |
|--------|-----|---------|-------|-------|--------|---------------|--------|------|
| (classes) | (47) | (10) | (43) | (10) | (45) | (196) | (397) | (10) |
| Linear Probe | **77.2±0.3** | 95.9±0.1 | 86.8±0.1 | 98.7±0.1 | **91.7±0.2** | **79.9±0.2** | 73.8±0.3 | 66.6±0.3 |
| Task Matrix | 75.7±0.5 | **96.2±0.4** | **87.2±0.3** | **99.03±0.1** | 89.1±0.6 | 79.7±0.5 | **74.8±0.3** | **66.7±0.7** |
| (best layer) | (11) | (6,8) | (11) | (7,8) | (11) | (11) | (11) | (8) |
| Fine-Tuned | 77.4±1.4 | 98.3±0.5 | 98.7±0.1 | 99.4±0.1 | 92.3±0.5 | 82.7±0.5 | 74.5±0.3 | 96.4±0.2 |

We also show results for all-MiniLM-L12-v2, DeIT, and DINOv3 in the Appendix, demonstrating our approach is generalizable across models [Wang et al., 2020, Touvron et al., 2020, Siméoni et al., 2025].

### 5.2 Task Matrices for Multitask Classification

We further investigate whether a *single* task matrix can exist for *multiple* datasets, as done by Ilharco et al. [2022] for model weight arithmetic. To formulate the task matrix for the multi-dataset domain, we replace our original linearity hypothesis with a joint assumption on linearity. Extending our original notion of concept representation, we instead posit that a transformation in model space can benefit multiple datasets. The task matrix then learns a joint mapping to an optimal space for all datasets. This means that the final layer embeddings $\hat{h}^L$ are sampled from a joint dataset $N = \{S_1, S_2, \ldots S_n\}$, while the base embeddings remain unchanged:

$$(h^i, \hat{h}^L) = (h^i, \bigcup_{S \in N} \hat{h}^L)$$

We then create task matrices following the technique outlined in Section 3.2. To evaluate task matrices across the selected datasets $\{d_1, ..., d_n\}$ on the test sample $j$, we multiply the same task

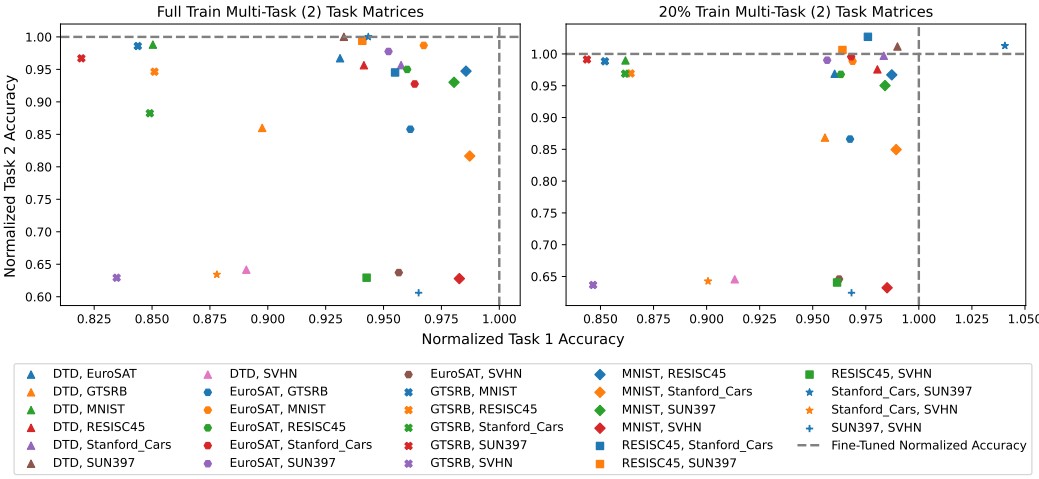

Figure 2: CLIP ViT-B/32 Vision 2 Task Augmentation. Learned linear approximations are beneficial for each dataset, and exhibit relative improvements in the data-scarce setting.

matrix $W^*$ with the intermediate representation $h^j$, and pass results through the respective fine-tuned classification heads $D_1, \dots D_n$ to obtain predictions.

As seen in Figure 2, which represents multi-task task matrices performance on all pairs of datasets $D_i \times D_j$, matrices maintain performance on both targeted tasks, validating the hypothesis above.

### 5.3 Ablation: Direct Readout from Base Model

One potential confounding factor with our methodology is determining whether task matrix performance arises from transformation or simply from the fine-tuned classifier head. To isolate these effects, we conducted a controlled ablation experiment testing the base model representations with a fine-tuned classifier head alone. As seen in the Appendix, the **Base w/ FT Classifier** method performs worse than task matrices on all datasets across all settings. By effectively replacing task matrices with the identity, the ablation demonstrates the necessity of the transformation for improved performance.

## 6 Conclusion

Recent results in interpretability suggest that models contain linear substructure, in particular under input-output constraints such as object prediction from relational examples. We apply the linear representation hypothesis beyond the constraints in previous work towards a broader application: the representational changes that result from gradient-based fine-tuning.

With this theoretical justification, we introduce task matrices as linear mappings between base and fine-tuned model states that improve the performance of a base model on specialized datasets across a wide range of tasks. We find that while performance varies in effectiveness across datasets, task matrices can often result in competitive performance with the specialized model itself. We observe further that these transformations can learn a range of tasks while retaining high individual accuracy, and that they are robust to reduced data regimes.

## Acknowledgments and Disclosure of Funding

We thank Ashwinee Panda, Gabe Grand, and Vasu Sharma for their valuable feedback, insightful discussions, and support throughout this research. We are grateful to Kevin Zhu and the Algoverse AI Research program for providing essential compute resources and logistical support that made this work possible.

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
