# OpenReview forum: "Linear Maps for Cross-Model Finetuning Transfer"
_NeurIPS.cc/2025/Workshop/UniReps — UniReps2025_

### Official Review · Reviewer_VmdL · 2025-09-08
**Intriguing idea, but insufficient empirical study**

**Confidence:** 3

**Review:**

The authors posit the "linearity assumption", the hypothesis that intermediary layer embeddings of pretrained models can be linearly mapped to fine-tuned representations of downstream tasks. They empirically show this "task matrix", obtained via least-squares to representations from the fine-tuned model, compares similarly to linear probing, and outperforming linear probing in a few instances. The writing is clear. The idea is intriguing, but the empirical study is not sufficient.

The point is to show that there is a linear transformation from pre-trained representations to fine-tuned ones. Therefore, using an FT head trained on the fine-tuned embeddings does not make sense as a baseline. Suitable baselines are rather linear probing as a lower bound, and fine-tuning as an upper bound. Especially Table 2 shows a significant gap to fine-tuning, suggesting the hypothesis does not hold in the text setup, while Table 1 shows insignificant improvements or even deterioration over linear probing in the vision domain, suggesting the "post-task matrix" embeddings are not more suitable for the downstream task. Furthermore, since a hyper-parameter scan over the layers has been performed for the task matrices, the same should also be done for at least the linear probe to eliminate the effect of picking a layer more aligned to the current task. It is unclear whether this has been accounted for.

**Score:**

2

**Topic Fit:**

3

---

### Official Review · Reviewer_zGVb · 2025-09-15

**Confidence:** 4

**Review:**

###  **Summary**

This paper investigates the relationship between pretrained embeddings and their fine-tuned counterparts, with a focus on testing the hypothesis that fine-tuned embeddings can be linearly approximated from pretrained ones. To this end, the authors learn a transformation matrix $W$ that maps pretrained embeddings to fine-tuned embeddings and evaluate whether such a linear transformation can (i) outperform linear probing and (ii) approach the performance of full fine-tuning.

### **Strengths**

The paper is clearly written and easy to follow, with the main ideas and contributions presented in a straightforward manner. The study of linear relationships in representation spaces is of broad interest across multiple research areas, and this work may inspire future research directions or valuable discussions within the community. Results suggest that linear encodings in transformer embedding spaces exist not only across layers within a model but also between pretrained and fine-tuned models. Beyond that, on certain datasets, the proposed linear mapping approach outperforms standard linear probes.

### **Weaknesses & Suggestions for Improvement**

- The current study is restricted to in-distribution fine-tuning scenarios regarding the pre-trained model. A natural extension would be to test whether the observed linear relationships holds under out-of-distribution (OOD) settings relative to the pretraining data in any degree. For example, one could investigate whether ImageNet-pretrained models fine-tuned on medical imaging datasets exhibit similar properties. Such experiments could potentially provide insights into whether the method could support better zero-shot performance.

- While the findings are intriguing, the paper would benefit from a deeper discussion of how this linear mapping relates to existing studies on representation alignment, transferability, or probing. This would help situate the contribution more clearly within the broader literature.

**Score:**

3

**Topic Fit:**

2

---

### Official Review · Reviewer_wJrm · 2025-09-16
**Review of "Task Matrices: Linear Maps for Cross-Model Finetuning Transfer across Modalities"**

**Confidence:** 5

**Review:**

This work proposes a novel framework for cross-model adaptation using task matrices: linear transformations learned between intermediate layers of a base model and the final layer of a fine-tuned model. The core hypothesis is that linear mappings can approximate fine-tuned behavior without retraining the entire model. This is intriguing and builds on prior work on linear representations in transformers. The authors validate this through extensive experiments across vision (CLIP, DeiT) and text (all-MiniLM, RoBERTa) models on 15 diverse datasets, showing competitive results in low-data regimes and ablation studies isolating the contribution of the task matrix itself.

# Strengths
1. While linear transformations for relational reasoning are well-studied, applying them to domain adaptation via fine-tuning is a fresh perspective. The paper effectively bridges interpretability research with practical adaptation techniques.
2.The breadth of experiments spanning image classification (DTD, EuroSAT), text tasks (HANS, Banking-77), and multimodal settings is a major strength. Tables 1-4 provide clear comparisons against baselines, and the inclusion of low-data settings (Tables 5-6, 9-10) adds practical relevance.
3, The analysis in Table 7-8 convincingly shows that the task matrix itself drives performance gains and not just the classifier head. This isolates the core contribution and strengthens the claim that linearities exist between base and fine-tuned states.
4.  The method requires only a single linear regression step making it simple and strong compared to full fine-tuning or adapter-based methods.

# Weaknesses
1. The paper does not compare task matrices against widely adopted techniques like LoRA, adapters, or prompt tuning. For example, LoRA achieves >90% of full fine-tuning performance with minimal overhead on vision tasks (Hu et al., 2021), yet this benchmark is absent. Without such comparisons, it is unclear whether task matrices offer a meaningful advantage beyond linear probes.
2. On text tasks (Table 2), task matrices underperform linear probes on Emotion (63.5% vs. 66.8%) and Banking-77 (88.3% vs. 90.9%). Similarly, on SVHN (Table 1), task matrices lag far behind fine-tuned performance (66.7% vs. 96.4%). The paper offers no analysis of why certain datasets (e.g., HANS, MNIST) benefit more than others e.g., is it due to dataset complexity, modality, or task structure? This limits generalizability.
3. The method requires access to a fine-tuned model to construct the task matrix. If a fine-tuned model is already available, why not use it directly? The paper never justifies the practical value of "distilling" fine-tuned knowledge into a linear transformation when the original fine-tuned model is accessible. This undermines the claimed "lightweight" advantage.

**Score:**

3

**Topic Fit:**

3